

# Atmospheric evolution of environmentally persistent free radicals in rural North China Plain: insights into water solubility and effects on PM2.5 oxidative potential

Xu Yang[1], Fobang Liu[1,*], Shuqi Yang[1], Yuling Yang[2], Yanan Wang[1], JingJing Li[2], Mingyu Zhao[2], Zhao Wang[1,3], Kai Wang[2], Chi He[1], Haijie Tong[4,*]

[1]Department of Environmental Science and Engineering, School of Energy and Power Engineering, Xi'an Jiaotong University, Xi'an, Shaanxi 710049, China

[2]State Key Laboratory of Nutrient Use and Management, College of Resources and Environmental Sciences, National Academy of Agriculture Green Development, National Observation and Research Station of Agriculture Green Development (Quzhou, Hebei), China Agricultural University, Beijing 100193, China

[3]Shaanxi Provincial Land Engineering Construction Group Co., Ltd., Xi'an, Shaanxi 710075, China.

[4]Institute of Surface Science, Helmholtz-Zentrum Hereon, Max-Planck-Str. 1, Geesthacht 21502, Germany

*Corresponding authors: Fobang Liu (fobang.liu@xjtu.edu.cn); Haijie Tong (haijie.tong@hereon.de)

## Abstract

Environmentally Persistent Free Radicals (EPFRs) represent a novel class of hazardous substances, posing risks to human health and the environment. In this study, we investigated the EPFRs in ambient fine, coarse, and total suspended particulate matter ($PM_{2.5}$, $PM_{10}$, TSP) in rural North China Plain, where local primary emissions of EPFRs were limited. We observed that the majority of EPFRs occurred in $PM_{2.5}$. Moreover, distinct seasonal patterns and higher g-factors of EPFRs were found compared to those in urban environments, suggesting unique characteristics of EPFRs in rural areas. The source apportionment analyses revealed atmospheric oxidation as the largest contributor (33.6%) to EPFRs. A large water-soluble fraction (35.2%) of EPFRs was determined, potentially resulting from the formation of more oxidized EPFRs through atmospheric oxidation processes during long-range/regional transport. Additionally, significant positive correlations were observed between EPFRs and the oxidative potential of water-soluble $PM_{2.5}$ measured by dithiothreitol-depletion and hydroxyl-generation assays, likely

attributable to the water-soluble fractions of EPFRs. Overall, our findings reveal the prevalence of water-soluble EPFRs in

rural areas and underscore atmospheric oxidation processes can modify their properties, such as increasing their water solubility.

This evolution may alter their roles in contributing to the oxidative potential of $PM_{2.5}$ and potentially also influence their impact

on climate-related cloud chemistry.

## 1. Introduction

Environmentally persistent free radicals (EPFRs) are a new class of risk substances that have garnered significant attention

in recent years (Yi et al., 2023; Vejerano et al., 2018). Unlike short-lived radicals, EPFRs are characterized by their stability,

with lifetimes ranging from days to months, and even indefinite (Gehling and Dellinger, 2013; Runberg et al., 2020). They

have been identified in various environmental matrices, including soil, sediment, leaves, industrial fly ash, household dust,

and atmospheric particulate matter (PM) (Jia et al., 2017; Vejerano and Ahn, 2023; Zhao et al., 2019b; Filippi et al., 2022;

Arangio et al., 2016). Of particular interest is their presence in inhalable atmospheric PM, which serves as a crucial carrier for

EPFRs, amplifying their detrimental health impacts. Toxicological studies have demonstrated that EPFRs can induce lung

damage by triggering oxidative stress in lung cells, primarily through the reactive oxygen species (ROS) generated from the

catalytic cycling of EPFRs (Wang et al., 2011; Balakrishna et al., 2009; Yi et al., 2023). Moreover, the persistent nature of

EPFRs and their activation potential by certain environmental factors (e.g., water molecules and ultraviolet light) enable them

to participate in diverse atmospheric reactions, initiating or propagating subsequent radical reactions (Truong et al., 2010;

Sarmiento and Majestic, 2023; Comandini et al., 2012). Thus, understanding the sources and properties of EPFRs is essential

for assessing their atmospheric and health impacts.

The occurrence of EPFRs was initially detected in aerosols originating from cigarette smoke (Pryor et al., 1983), and later

also in combustion-derived particles as well as ambient PM (Dalal et al., 1991). Various combustion and thermal processes

including industrial processes, coal combustion, biomass burning, engine exhaust, and waste incineration, have been identified

as important sources of EPFRs (Liu et al., 2021; Yang et al., 2017; Wang et al., 2020a). Mechanistic studies suggest that EPFRs

can be generated and stabilized on the surfaces of transition metal-doped particles in the post-flame and cool-zone regions of

combustion systems (Cormier et al., 2006). In addition, secondary chemical processes of organic molecules, may also

contribute to the presence of EPFRs in atmospheric PM (Chen et al., 2019; Wang et al., 2020a). Previous work has shown that

EPFRs can be formed through the oxidation of organic molecules by ozone and photochemical reactions of PM (Borrowman

et al., 2016; Sarmiento and Majestic, 2023; Qin et al., 2021; Tong et al., 2018). Various factors, including solar irradiation,

humidity, and the types of precursors, may influence the formation of secondary EPFRs (Sarmiento and Majestic, 2023; Chen

et al., 2019; Liu et al., 2023).

    Multiple studies conducted in recent decades have investigated the composition of EPFRs in atmospheric PM (Yang et

al., 2017; Arangio et al., 2016; Xu et al., 2020). The identified types of EPFRs include carbon-centered and oxygen-centered

free radicals, as well as carbon-centered free radicals with a nearby heteroatom. These radicals often manifest as

cyclopentadienyl, semiquinone, and phenoxyl radicals (Ai et al., 2023). Notably, Chen et al. (2018b) revealed that the dominant

fraction of EPFRs existed within nonsolvent-extractable organic matter of urban $PM_{2.5}$, underscoring the need for further

exploration into the organic molecules associated with ambient EPFRs.

    EPFRs exhibit redox activity, capable of reducing oxygen and facilitating the formation of ROS such as hydroxyl radicals

($\bullet$OH) and superoxide radicals ($O_2^-\bullet$) (Hwang et al., 2021; Guo et al., 2020). Consequently, EPFRs may serve as crucial

hazardous components contributing to the toxicity of atmospheric PM. Li et al. (2023) found that EPFRs can contribute to the

oxidative toxicity of both water-soluble and -insoluble fractions of atmospheric PM. The types of EPFRs and their extractability

may influence their roles in ROS formation (Zhao et al., 2019b). Moreover, ambient EPFRs have been detected in both fine

and coarse particles, with observed seasonal and spatial variations in their size distribution (Jia et al., 2023; Wang et al., 2022).

The presence of EPFRs in different particle sizes may pose various health risks to humans due to differences in deposition

efficiency within the respiratory tract.

The investigations of airborne EPFRs in urban areas, heavily influenced by traffic, industrial, and residential emissions,

have been the primary focus of previous studies (Yang et al., 2017; Wang et al., 2020a). However, EPFRs, characterized by

their long lifetimes, can undergo transport over considerable distances and reach rural areas with minimal local emissions.

Indeed, the long-range transport of EPFRs has been demonstrated (Chen et al., 2018a). During the transport, the characteristics

of EPFRs may undergo evolution through atmospheric chemical processes, potentially altering their roles in ROS formation.

Despite this, investigations into the characteristics of airborne EPFRs in areas with limited local emissions remain sparse.

Insights into airborne EPFRs in such areas will allow a better understanding of the atmospheric transformation and fate of

EPFRs as well as their atmospheric and health effects.

Therefore, to enhance our understanding of EPFR evolution during atmospheric transport and its effects on ROS

formation by the corresponding PM, we collected yearlong PM samples in a typical rural area located in the North China Plain

(NCP), where local combustion emissions contributing to EPFRs are minimal. On the other hand, northern China, including

NCP, is one of the most polluted regions in China, characterized by significant combustion sources, making the selected

location ideal for our research objectives. Specifically, our study aims to (i) investigate the characteristics of EPFRs, including

their concentration, size distribution, and seasonal variations in the studied region; (ii) determine the sources of EPFRs; (iii)

explore the roles of EPFRs' speciation in contributing to the oxidative potential of PM.

## 2. Methods

### 2.1 PM sample collection

Ambient PM samples were collected at a rural site (36°51'48''N 115°00'58''E) in Quzhou county in the NCP from April

2022 to March 2023. The sampling site represents a typical rural environment, predominantly surrounded by croplands and

devoid of significant local industrial sources (Figure S1). Sequential 24-hour fine (PM$_{2.5}$), coarse (PM$_{10}$), and total suspended

particles (TSP) were collected using a high-volume sampler (TH-1000CII, Tianhong, China) at a flow rate of 1.05 m$^3$/min.

The PM samples were collected onto prebaked (900 °C) quartz filters (Munktell, type MK360) and stored at -20 °C until

analysis. In total, ninety-five PM samples were collected during the whole sampling. A summary of the number of PM$_{2.5}$, PM$_{10}$,

and TSP samples collected in each season is provided in Table S1. In addition, other air quality parameters (SO$_2$, NO$_2$, O$_3$, and

CO) data were obtained from the monitoring site nearest to the sampling site from the local environmental monitoring center.

**2.2. EPFRs analysis**

Three punches (1.2 cm$^2$ per punch) of each filter were inserted into a quartz tube (5 mm I.D., SP Wilmad-LabGlass) for

EPFR measurements using an EPR spectrometer (A300-9.5/12, Bruker). The detection parameters for EPFRs were set as

follows: a modulation frequency of 100 kHz; a microwave frequency of 9.8485 GHz; a microwave power of 1.76 mW; a

modulation amplitude of 1.00 G; a sweep width of 150 G; a time constant of 81.92 ms; a receiver gain of 1 × 10$^3$ G. All EPR

measurements were conducted at room temperature. To minimize noise in EPR signals, a baseline correction was performed,

followed by fitting the signals using a Gaussian function via the least squares method. EPFR concentrations for the filters were

determined by comparing the peak area with a calibration curve (Figure S2) generated using a common radical stand, 4-

hydroxy-2,2,6,6-tetramethylpiperidin-1-oxyl (TEMPOL) (Arangio et al., 2016).

In addition, we conducted water extraction and acidification experiments on the samples to determine the reduction of

EPFR contents on the filter samples after treatment. The procedures of water extraction and acidification were adopted from

previous studies (Chen et al., 2018b; Yang et al., 2017). For water extraction, all PM$_{2.5}$ samples were processed alongside two

randomly selected PM$_{10}$ and TSP samples from each season. Briefly, three punches (1.2 cm$^2$ per punch) of each filter were

immersed in 3 mL of deionized water for 14 hours under dark condition. Regarding acidification, two PM$_{2.5}$, PM$_{10}$, and TSP

samples in each season were randomly selected, and two punches (1.2 cm$^2$ per punch) of each filter were immersed in 2 mL of

6 M HCl solution for one hour under dark condition. Subsequently, the filter punches were dried by a vacuum freeze dryer

before conducting EPFRs measurements using the aforementioned detection parameters.

## 2.3 Carbonaceous fractions and element analyses

A 1.0 cm$^2$ punch of each sample filter was analyzed for organic carbon (OC) and elemental carbon (EC) following the

Interagency Monitoring of Protected Visual Environments (IMPROVE) thermal/optical reflectance (TOR) protocol using the

DRI Model 2001 carbon analyzer. Different carbonaceous fractions, including OC1–4 and EC1–3, were isolated and quantified

based on their thermal stability.

Twelve elements (Li, Mg, Al, Si, K, Ca, Cr, Mn, Fe, Cu, Zn, and Pb) were analyzed using inductively coupled plasma

mass spectrometry (ICP-MS, Thermo Scientific iCAP RQ). Prior to analysis, a 3.6 cm$^2$ punch of each filter was digested using

1 mL of aqua regia (HNO$_3$+3HCl, $v$:$v$) at 99 °C and a rotational frequency of 350 rpm for 24 hours. After digestion, the extracts

were filtered through a 0.22 µm PTFE syringe filter and then diluted to 5 mL deionized water with 2% HNO$_3$.

## 2.4 Oxidative potential measurements

The oxidative potential of PM samples was measured by two techniques, the dithiothreitol (DTT) depletion assay and

•OH production assay. In the DTT assay, the decay of 100 µM DTT by PM in phosphate buffer was monitored over a 40-

minute incubation at 37 °C. The rest DTT after the incubation was quantified by its reaction with dithiodinitrobenzoic acid,

yielding ultraviolet-detectable 2-nitro-5-thiobenzoic acid. In the •OH production assay, the terephthalate (10 mM) in phosphate

buffer was used to measure •OH radical formation by PM throughout 2-hour incubation at 37 °C. At pH 7.4, terephthalate

reacted with •OH to form stable and highly fluorescent hydroxyterephthalic acid (2-OHTA), and the production rate of •OH

was calculated based on the produced 2-OHTA, as the formation of 2-OHTA is proportional to the generation of •OH. Note

that samples were incubated at a PM concentration of 100 μg/mL for both assays. Meanwhile, both total OP (Total-OP) and water-soluble OP (WS-OP) were determined in this work. For total OP determination, unfiltered PM extracts with filter punches left in the extracts were directly incubated with the probes. While for water-soluble OP, the extract was filtered through a 0.22 μm PTFE syringe filter before incubating with probes. The OP contribution from water-insoluble PM

components (water-insoluble OP, WIS-OP) was considered as the difference between Total-OP and WS-OP. A detailed description of the DTT and •OH assays can be found in the Supporting Information (Text S1).

## 2.5 Statistical analysis

### 2.5.1 Positive matrix factorization

The EPA positive matrix factorization (PMF 5.0) model was employed to apportion the sources of EPFRs and PM in this

study. The PMF model is an advanced multivariate factor analysis tool widely utilized for source apportionment of environmental pollutants (Heo et al., 2013; Wang et al., 2019). The input data include the concentrations and uncertainties of PM, EPFRs, organic carbon fractions (OC1, OC2, OC3), elemental carbon fractions (EC1, EC2, EC3), $SO_2$, $NO_2$, CO, $O_3$, and the twelve elements. The uncertainties of each variable were calculated using the equation: Uncertainty = K × Concentration, where K denotes analytical uncertainty (Wang et al., 2019). For PM, EPFRs, OC, and EC, K was set as 10%(Jang et al., 2020).

For metal elements and meteorological parameters, K was set as 15% (Ikemori et al., 2021). Missing values and associated uncertainties were estimated by substituting the median concentrations of the components and four times the median value of the components to mitigate their impact on the results (Wang et al., 2019).

The PMF model was run with four to seven factors and with random seeds. The six-factor result was considered as the optimal one based on minimal Q value (indicating the variation between observation and the model is the least), $Q_{Robust}/Q_{Theo}$

( < 2), scaled residuals (within ± 3), and the interpretable profiles from the literature (Reff et al., 2007; Ramadan et al., 2000). Bootstrap and displacement analyses were conducted to estimate the uncertainty of the PMF model with six factors (Brown et

al., 2015), and the results are shown in Figure S3. The bootstrap factor mapping exceeded 93% for all factors without any

displacement run exchanges.

### 2.5.2 Correlation analysis

Correlation analysis was performed using Pearson's correlation coefficients and two-tailed significance tests by IBM

SPSS Statistics 27.

### 2.6 Backward trajectories

To characterize the origins and transport pathways of the air masses to the sampling site, 48-h backward trajectories of

the air masses were simulated using the Hybrid Single Particle Lagrangian Integrated Trajectory (HYSPLIT) model. The input

meteorological data were acquired from the Global Data Assimilation System (ftp://arlftp.arlhq.noaa.gov/pub/archives/gdas1).

## 3. Results and discussion

### 3.1 Characteristics of EPFRs in different sizes of PM

Figure 1a illustrates the box plots of volume-normalized (EPFRv) concentrations of EPFRs in different sizes of PM.

EPFRv levels in $PM_{2.5}$, averaging $(5.6 \pm 1.1) \times 10^{12}$ spins/m$^3$, accounted for over 95.2% of those in $PM_{10}$ $((5.8 \pm 1.0) \times 10^{12}$

spins/m$^3$) and TSP $((5.9 \pm 1.1) \times 10^{12}$ spins/m$^3$). However, the average mass concentration of $PM_{2.5}$ only represented 61.8% of

that in $PM_{10}$, and 47.5% of that in TSP (Figure S4). These results suggest that the majority of airborne EPFRs are present in

$PM_{2.5}$, with only a small portion occurring in the 2.5-10 μm size range.

This is further evident by the results of mass-normalized (EPFRm) concentration of EPFRs, as depicted in Figure 1b. The

average EPFRm in $PM_{2.5}$ $(6.5 \pm 3.5) \times 10^{16}$ spins/g) was approximately 1.7 times that in $PM_{10}$ $((3.9 \pm 2.1) \times 10^{16}$ spins/g), and

2.2 times that in TSP $((3.0 \pm 1.7) \times 10^{16}$ spins/g). Consistent with our finding, previous studies have also observed a

predominant fraction of EPFRs in smaller sizes (< 2.5 μm) of PM (Chen et al., 2020; Dugas et al., 2016). The strong association

of EPFRs with fine particles may stem from multiple factors. Firstly, airborne EFPRs primarily originate from the combustion

of organic materials and multiphase chemistry of gaseous organics in the atmosphere, processes that predominantly form fine

particles (Tian et al., 2009; Arangio et al., 2016). Additionally, the larger specific surface area and porous structure of fine

particles facilitate the attachment of EPFRs, leading to their higher retention within this fraction (Yang et al., 2017).

Table S2 presents a summary of the concentrations of EPFRv and EPFRm in this study compared to existing literature.

Both EPFRv and EPFRm in this study were lower than in most urban and suburban environments (Wang et al., 2019; Jia et

al., 2023; Yang et al., 2017). This implies that there were only limited local sources directly emitting EPFRs at this rural site.

This is further supported by the seasonal variation of EPFRs. Notably, higher EPFRv levels were observed in summer than in

winter (Figure 1c), with the highest EPFRm occurring during summer (Figure 1d) in the studied area. This contrasts with

findings from previous studies, which indicated that winter typically exhibits the highest EPFRv, especially for cities in

northern China with prominent local emission sources of EPFRs, such as large-scale coal consumption (Wang et al., 2019; Jia

et al., 2023; Ai et al., 2023). The distinct seasonal characteristic of EPFRs observed in our study could be attributed to the

limited residential and industrial activities in rural NCP. On the other hand, the highest EPFRm levels in summer may be linked

to enhanced atmospheric oxidation, promoting EPFR formation due to elevated concentrations of photo-oxidants and stronger

solar irradiation (Borrowman et al., 2016; Sarmiento and Majestic, 2023; Shiraiwa et al., 2011).

In addition to examining the concentrations of EPFRs, we also investigated the g-factors of EPFRs across different sizes

of PM, as the g-factor is an important indicator of EPFRs characteristics. We observed that the g-factors for all PM samples

ranged between 2.0033 and 2.0037, with a mean value of 2.0035 ± 0.0001 (Figure S5). These g factors are higher than the

values obtained from PM samples collected from highway and urban locations (Hwang et al., 2021; Fang et al., 2023). A g-

factor below 2.003 signifies the presence of carbon-centered radicals (e.g., cyclopentadienyl), while a g-factor exceeding 2.004



indicates oxygen-centered radicals (e.g., phenoxy, semiquinone radicals). A g factor falling between 2.003–2.004 suggests a carbon-centered radical with adjacent oxygen atoms (Ai et al., 2023; Yang et al., 2017). The observed g factors in this work indicate the presence of carbon-centered radicals with adjacent oxygen atoms and oxygen-centered radicals. Furthermore, we

noted a decreasing trend in the g-factor with particle size (i.e., $PM_{2.5} > PM_{10} > TSP$), indicating closer proximity of the unpaired electron to the oxygen center in smaller particles. This trend could be attributed to the increased exposure of porous structures in smaller particles, rendering them more susceptible to oxidation (Yang et al., 2017).

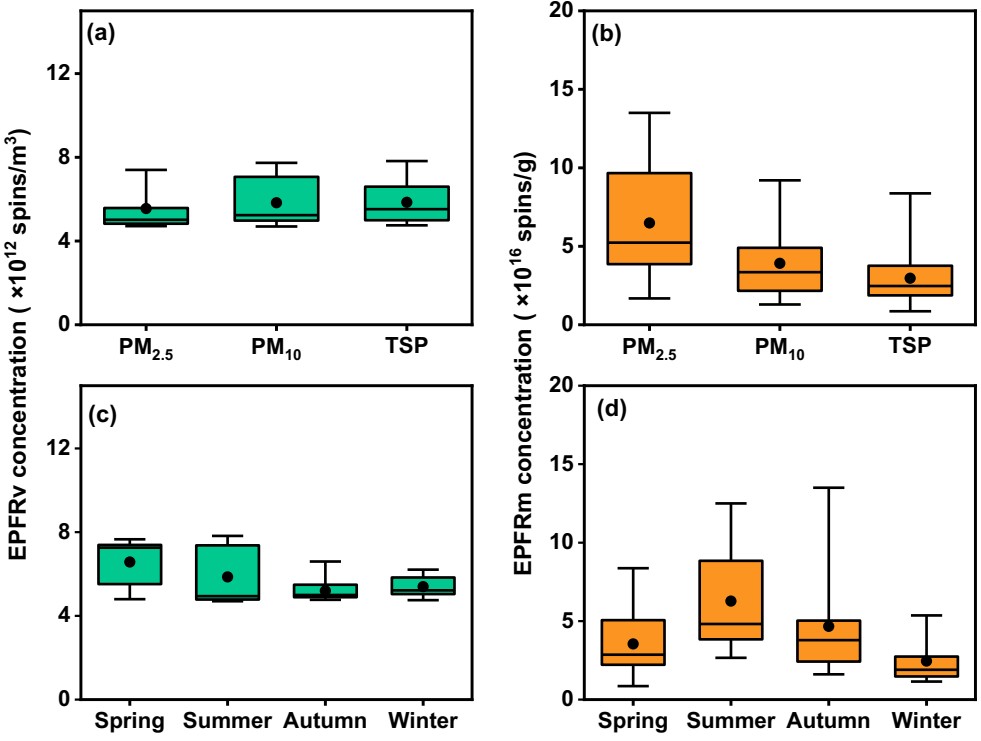

**Figure 1.** The concentrations of EPFRv (a and c) and EPFRm (b and d) in different sizes of PM samples and different

seasons. The boxes represent the 25th percentile (lower edge), median (solid line), mean (solid dot), and 75th percentile (upper edge). The whiskers represent the minimum and maximum.

**3.2 Sources of EPFRs**

The PMF analysis was utilized to identify and quantify the sources contributing to EPFRs. As shown in Figure 2a, six

primary source factors were determined. Factor 1 was attributed to atmospheric oxidation, characterized by its elevated

proportion of $O_3$. The presence of $O_3$ signifies atmospheric oxidative capability (Ainur et al., 2023; Wang et al., 2020b). Factor

2 exhibited high proportions of Al, Fe, Cr, $SO_2$, CO, and EC, suggesting industrial emissions. Al and Fe could be associated

with iron and steel production and the metal industry (Khobragade and Ahirwar, 2022). The availability of Cr is linked to fossil

fuel combustion or oil combustion (Alleman et al., 2010; Begum et al., 2011). $SO_2$ and CO are typical tracers of coal

combustion (Johnson et al., 2006; Kundu et al., 2010; Wang et al., 2019), and they (and also EC) could be emitted from coal-

based industries. Factor 3, characterized by high proportions of $SO_2$, $NO_2$, and CO, was identified as coal combustion (Johnson

et al., 2006; Kundu et al., 2010; Wang et al., 2019). Previous research indicates that $NO_2$ emissions, although often associated

with vehicle emissions, are also prevalent in coal combustion (Ainur et al., 2023; Lei et al., 2016; Wang et al., 2018a) .In factor

4, high proportions of OC3, OC4, EC1, EC2, EC3, Mn, Cu, and Pb were observed, suggesting motor vehicle emissions. EC1,

OC2, OC3, and OC4 primarily originate from petrol vehicle emissions, while EC2 and EC3 are closely associated with diesel

vehicle emissions (Kim and Hopke, 2004; Ai et al., 2023). Additionally, Mn (from unleaded gasoline additives), Pb (from

gasoline additives), and Cu (from brake linings) contribute to this factor (Sharma et al., 2014). Factor 5 displayed a high

loading of OC1 and a considerable loading of OC2, indicative of biomass burning. OC1 and OC2 have been linked to biomass

combustion in previous studies (Stanimirova et al., 2023; Cao et al., 2005; Dong et al., 2022). Factor 6 exhibited high

proportions of Mg and Ca, suggesting soil dust as these elements are major constituents of the Earth's crust (An et al., 2015;

Liu et al., 2022b).

In summary, the six primary sources contributing to EPFRs were identified as atmospheric oxidation, industrial emissions,

coal combustion, motor vehicle emissions, biomass burning, and soil dust. As shown in Figure 2b, atmospheric oxidation

emerged as the largest contributor (33.6%) to EPFRs during the entire sampling period. The formation of EPFRs through

atmospheric oxidation could occur via photo-oxidant oxidation (reactions of polycyclic aromatic hydrocarbons (PAHs) with



solar radiation, $O_3$ and •OH) and metal-catalyzed redox reaction (PAHs interacting with metals under visible and UV light

irradiation). Solar radiation and $O_3$ can directly generate EPFRs by disrupting the stable covalent bonds of organic substances,

while metals act as electron acceptors, facilitating electron transfer with PAHs to form EPFRs, as suggested by previous studies

(Borrowman et al., 2016; Shiraiwa et al., 2011; Chen et al., 2019; Wang et al., 2020a).

The significant contribution of atmospheric oxidation observed in our study contrasts with findings from urban areas,

where primary sources typically dominate airborne EPFRs (Ainur et al., 2023; Ainur et al., 2022). The results of seasonal

source apportionments (Figure 2b) further illustrate much higher contributions of atmospheric oxidation in spring (42.2%) and

summer (50.3%) compared to autumn (21.3%) and winter (18.6%). The elevated contribution in summer aligned with the

expectation of stronger photochemistry. The relatively high contribution in spring could be attributed to the dominance of long-

range transport of air mass (60%, Figure S6a), which was markedly higher than the proportions (< 23%) observed in the other

seasons. Previous studies have suggested that long-range transport of air mass favors atmospheric oxidation occurrence more

than the air mass from regional/short-range transport (Ramya et al., 2023; Zhong et al., 2022).

Industrial emissions (30.8%) were the second largest contributor to annual EPFRs, followed by coal combustion (14.5%),

motor vehicle emissions (10.3%), biomass burning (6.6%), and soil dust (4.2%). Industrial emissions and coal combustion

have been widely suggested as significant sources of EPFRs (Wang et al., 2020a; Yang et al., 2017; Wang et al., 2018b).

Although industrial activities were limited at this rural site, emissions might originate from surrounding industrial cities such

as Handan in the south, as indicated by backward trajectory analysis (Figure S6). Over 17% of air mass was identified as

regional transport from the south, particularly notable in autumn (54%), corresponding to a higher contribution (44.7%) of

industrial emissions to EPFRs compared to other seasons. Winter exhibited notably higher contributions from coal combustion

and biomass burning, as coal and biomass are primary fuels for residential heating in rural NCP during this season.

While motor vehicle emissions accounted for a small contribution to EPFRs, they constituted the largest contributor



(43.8%) to PM (Figure S7), suggesting differing components contribute to EPFRs and PM. For instance, volatile organic

compounds (e.g., toluene) and nitrogen oxides emitted from vehicle exhausts may significantly contribute to PM formation

but are generally not considered as major contributors to EPFRs (Nagpure et al., 2016; Gao et al., 2022; Wang et al., 2019).

Regarding soil dust, recent studies indicate that PAHs readily adsorb on mineral surfaces, likely forming "cation-π" interactions

with active sites. This interaction promotes electron transfer from aromatic compounds to surface cations on clay surfaces,

facilitating the formation of intermediate radicals/final products (Zhao et al., 2019a; Ni et al., 2023).

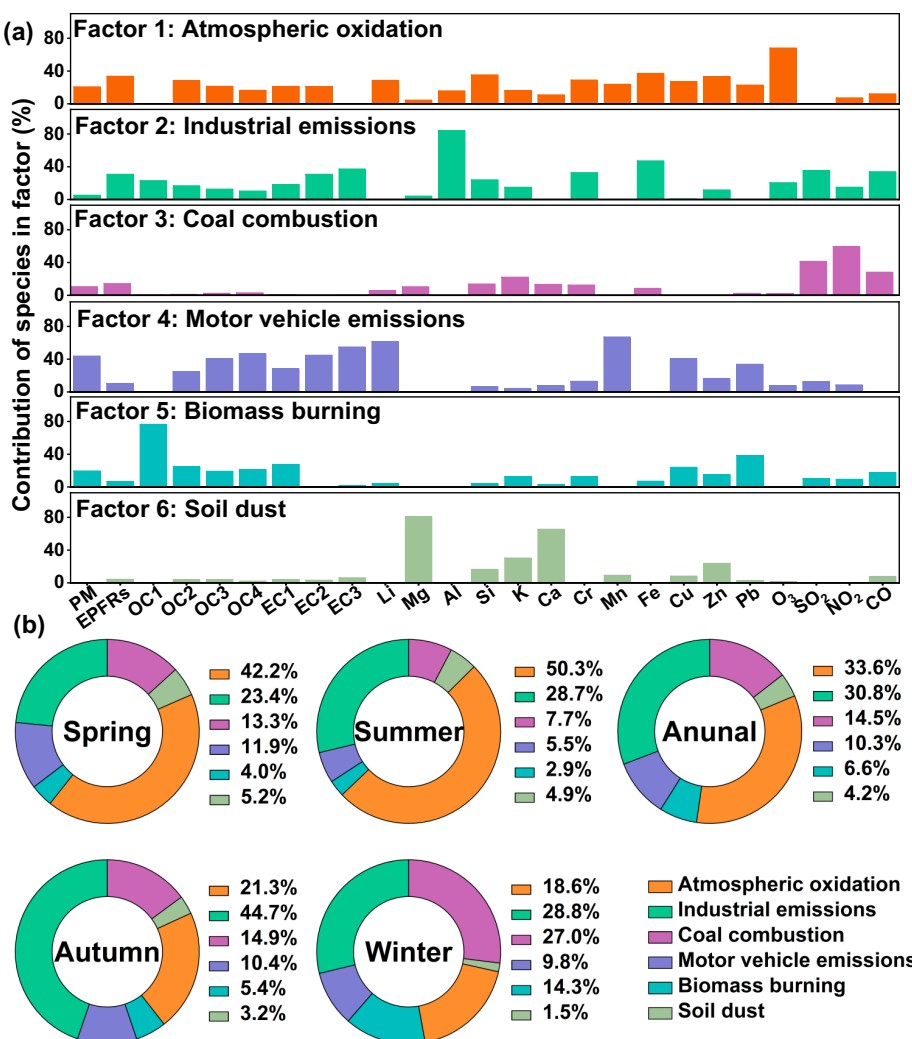

**Figure 2.** (a) Factor profile obtained by positive matrix factorization analysis. (b) Seasonal and annual contributions of the six factors to



EPFRs.

## 3.3 Associations of EPFRs with OP of PM

### 3.3.1 OP of PM

Oxidative potential, defined as the catalytic generation of ROS by PM components, serves as a plausible metric for assessing PM toxicity (Abrams et al., 2017; Weichenthal et al., 2016; Daellenbach et al., 2020; Bhattu et al., 2024). EPFRs have been identified as significant contributors to the OP of PM due to their ability to catalytically generate ROS (Gehling et al., 2014; Hwang et al., 2021). In this work, we also measured the OP of the PM samples using two commonly used assays: dithiothreitol-depletion ($OP^{DTT}$) and hydroxyl-generation ($OP^{\bullet OH}$) assays. $OP^{DTT}$ is found to be a good indicator of the production of $O_2^-$ and $H_2O_2$ but does not capture •OH generation (Fang et al., 2019). Thus, the combined application of two assays provides complementary insights into the role of EPFRs in ROS generation. Further, WS-OP, WIS-OP, and Total-OP were all determined to explore their potential correlations with EPFRs.

Tables S3 and S4 summarize the OP results in this work. A detailed discussion of OP values between the present work and literature can also be seen in the Supporting Information (Text S2). In short, consistent with the findings of EPFRm, the OPm values in this work were also lower than in most urban and suburban environments. This disparity suggests fewer redox-active PM components in the studied rural area, likely due to the absence of significant local emission sources of pollutants.

Figure 3a illustrates the box plots of mass-normalized WS-OP, WIS-OP, and Total-OP determined by the two assays. Generally, Total-OP levels were higher than WS-OP and WIS-OP, demonstrating measurable contributions of both water-soluble and -insoluble species to the overall OP of PM. Additionally, WS-OP accounted for a larger fraction of Total-OP in both assays (WS-$OP^{DTT}$: $67.8 \pm 20.5\%$; WS-$OP^{\bullet OH}$: $56.1 \pm 22.6\%$). Furthermore, a reverse relationship between particle size and the contribution of WS-OP to Total-OP was observed, with the largest contribution in $PM_{2.5}$, followed by $PM_{10}$ and TSP (Figure 3b). A significantly higher contribution of WS-$OP^{DTT}$ to Total-$OP^{DTT}$ has also been observed for ambient $PM_{2.5}$ at




multiple locations worldwide (Gao et al., 2017; Yang et al., 2024; Li et al., 2024). Yet, PM samples near highways and road

dust have shown a higher fraction of WIS-OP$^{DTT}$ than WS-OP$^{DTT}$ (Zhang et al., 2024; Li et al., 2023). This could be due to a

lower solubility of PM at the sites near primary emissions. These divergent results suggest that atmospheric oxidation processes

may alter the role of PM components in contributing to OP.

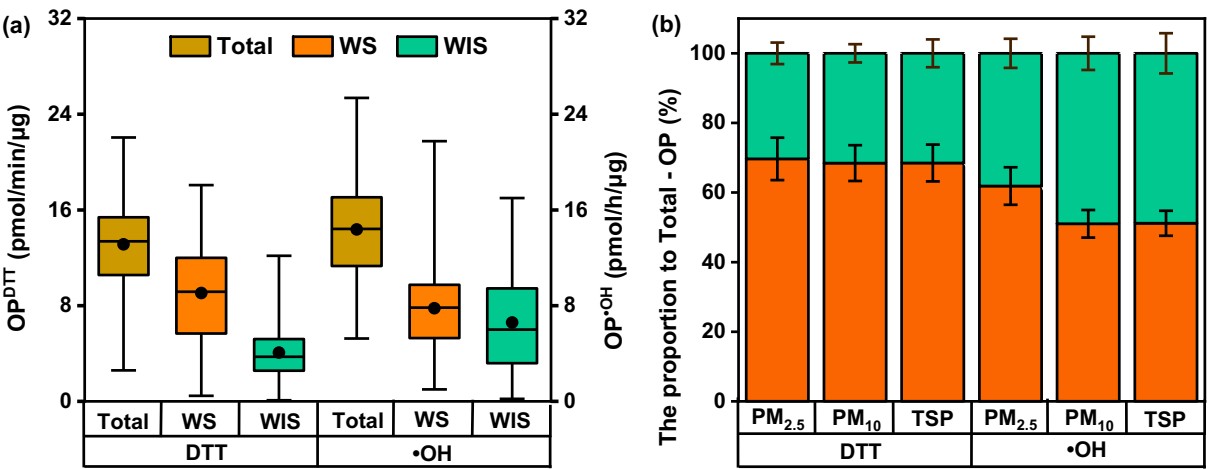

**Figure 3.** (a) The concentrations of total, water-soluble (WS), water-insoluble (WIS) fractions of OP$^{DTT}$ and OP$^{•OH}$. The boxes represent the 25$^{th}$ percentile (lower edge), median (solid line), mean (solid dot), and 75$^{th}$ percentile (upper edge). The whiskers represent the minimum and maximum. (b) Proportions of WS-OP and WIS-OP to Total-OP in different sizes of PM samples and the bar indicates the standard error.

### 3.3.2 Associations among EPFRs, OP, and PM components

Figure 4 presents the results of correlation analyses between OP$^{DTT/•OH}$ and the determined chemical species, respectively,

and the detailed information is also listed in Tables S5–7. Notably, the chemical species associated with Total-OP and WS-OP

exhibited similarities, particularly for PM$_{2.5}$. This may be partly because Total-OP was primarily comprised of WS-OP (Figure

3b). Also, good correlations were observed between Total-OP and WS-OP, except for the OP$^{•OH}$ of TSP samples (Figure S8).

In addition, we observed that OP$^{DTT}$ and OP$^{•OH}$ were sensitive to different chemical species (Figure 4). Specifically, OC, EC,

EPFRs, Fe, and Cr showed good correlations with Total-OP$^{DTT}$ and WS-OP$^{DTT}$; while Cu exhibited a good correlation with

Total-OP$^{•OH}$ and WS-OP$^{•OH}$ ($r > 0.5$, $p < 0.05$). As aforementioned, OP$^{DTT}$ is a good indicator of the production of O$_2^-$ and H$_2$O$_2$



(Fang et al., 2019). Thus, these results indicate that different chemical species contribute to the production of distinct types of ROS (Campbell et al., 2021; Jin et al., 2019; Calas et al., 2018). In terms of WIS-OP, only OC showed a moderate correlation ($r = 0.43$, $p < 0.05$) with WIS-OP$^{DTT}$. No species displayed a positively good correlation with WIS-OP$^{•OH}$, implying that WIS-OP might arise from complex synergistic/antagonistic interactions among PM components (Charrier and Anastasio, 2015; Yu

et al., 2018). However, further research is warranted to elucidate the mechanisms and PM components responsible for WIS-OP.

Interestingly, EPFRs in PM$_{2.5}$ exhibited significant correlations with WS-OP but not with WIS-OP (Figure 4). This observation contradicts previous studies reporting that the majority of EPFRs are not water-extractable (Chen et al., 2018b; Guo et al., 2023; Wang et al., 2018b). Given the source apportionment results showing atmospheric oxidation as the largest

contributor to EPFRs, we hypothesized that atmospheric oxidation processes may have increased the water solubility of rural EPFRs, leading to the observed significant correlations.





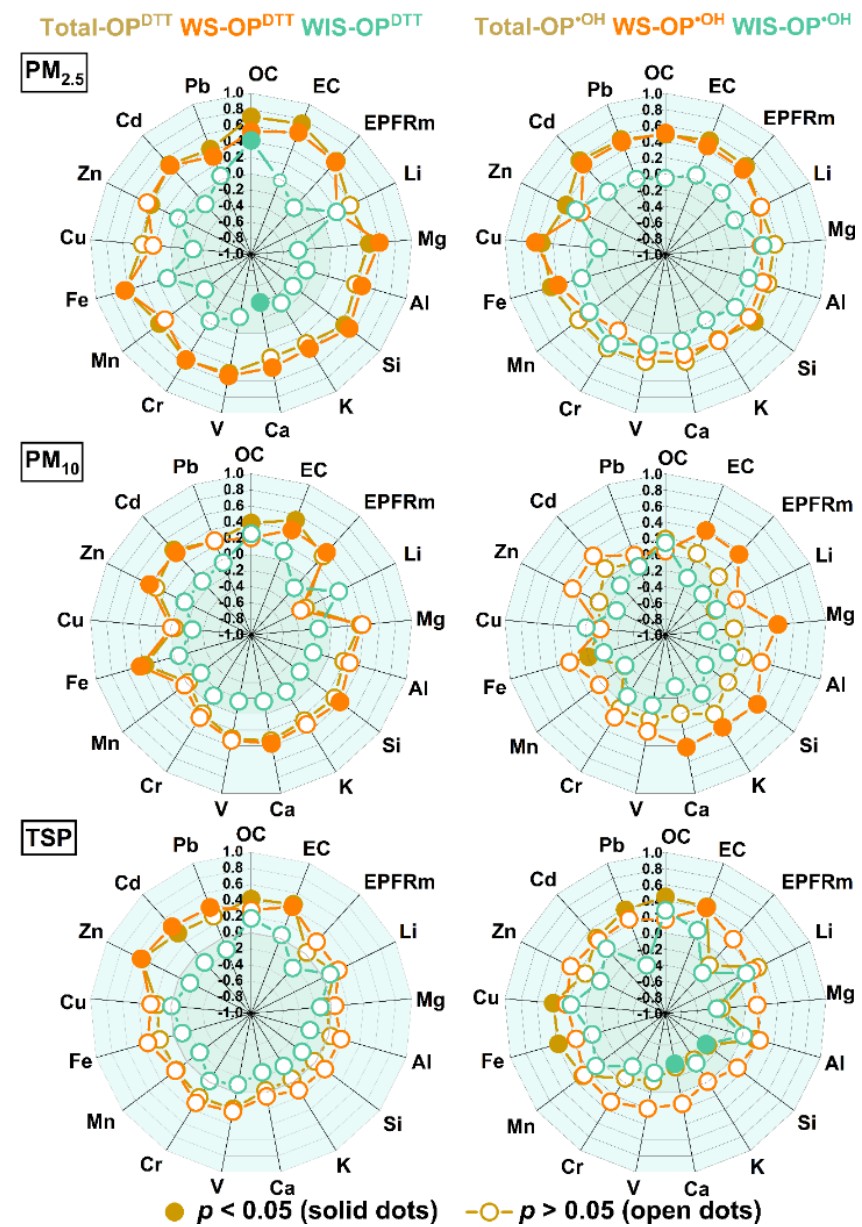

**Figure 4.** Correlation coefficients (Pearson's r) of mass-normalized OP (Total/WS/WIS) with ambient concentrations of selected PM components.

### 3.3.3 Solubility of EPFRs and its linkages with OP

To demonstrate the possibly high solubility of EPFRs, additional experiments were conducted with PM filters extracted

by water prior to EPFRs analysis. On average, a 35.2% reduction in EPR signals was observed after water extraction (Figure 5), indicating that the same percentage of EPFRs was water-soluble (A detailed water-soluble fraction of EPFRs in each filter is listed in Table S8). This result is substantially higher than the reported water-soluble fraction (0.2%) of EPFRs in urban

PM$_{2.5}$ of Xi'an (Figure 5b), where the majority of EPFRs likely consisted of graphene oxide analogues (Chen et al., 2018b). In addition, the result is also higher than that (11%) of EPFRs emitted from biomass burning (Guo et al., 2023), suggesting a high water-soluble fraction of EPFRs in the studied rural area.

It is widely acknowledged that EPFRs from combustion processes are formed through the electron transfer mediated by transition metals on organic combustion byproducts such as PAHs (Liu et al., 2022a; Vejerano et al., 2018). The EPR signal of

these metal-EPFRs complexes can be substantially reduced ($>90\%$, Figure 5c) after acidification, as demonstrated by previous work (Guo et al., 2023). We also conducted the same acidification procedure and observed, on average, a 70% reduction in EPR signals (Figure 5c, detailed information is provided in Table S9). Although the reduction in our work is not as substantial as that of biomass-burning particles (Guo et al., 2023), the result indicates that there existed a large fraction of EPFRs in the form of metal-EPFRs complexes, likely originating from combustion sources (e.g., coal and biomass combustions). In addition,

EPFRs showed significantly good correlations (r > 0.5, $p$ < 0.01; Table S10) with certain transition metals species (i.e., Fe, V, Zn, Cr), suggesting their formation was likely associated with transition metals. The higher water-soluble fraction of EPFRs in our work compared to that of biomass-burning particles could be relevant to the increased polarity of EPFRs by atmospheric oxidation processes (i.e., the aging of metal-EPFRs complexes). It is also worth noting that the reduction of EPR signals is at a much greater extent than that (24%, Figure 5c) of EPFRs (mostly consisting of graphene oxide analogues) in urban PM$_{2.5}$ of

Xi'an (Chen et al., 2018b).

In addition to the potentially oxidized/aged metal-EPFRs complexes, another significant source of water-soluble EPFRs could be EPFRs directly generated by secondary chemical processes. Chen et al. (2019) found that water-extracted humic-like



substances were the primary precursors of secondary EPFRs formed by visible-light illumination in ambient PM. A laboratory study by Tong et al. (2018) demonstrated that secondary organic aerosol formed by photooxidation of naphthalene contained

EPFRs at levels comparable to those found in ambient PM. Given that atmospheric oxidation was the main source of EPFRs and PM in our studied region, these secondary chemical processes likely had a substantial impact on ambient PM, contributing to the formation of secondary EPFRs. However, further research is needed to fully elucidate and quantify the contribution of secondary formation pathways to water-soluble EPFRs.



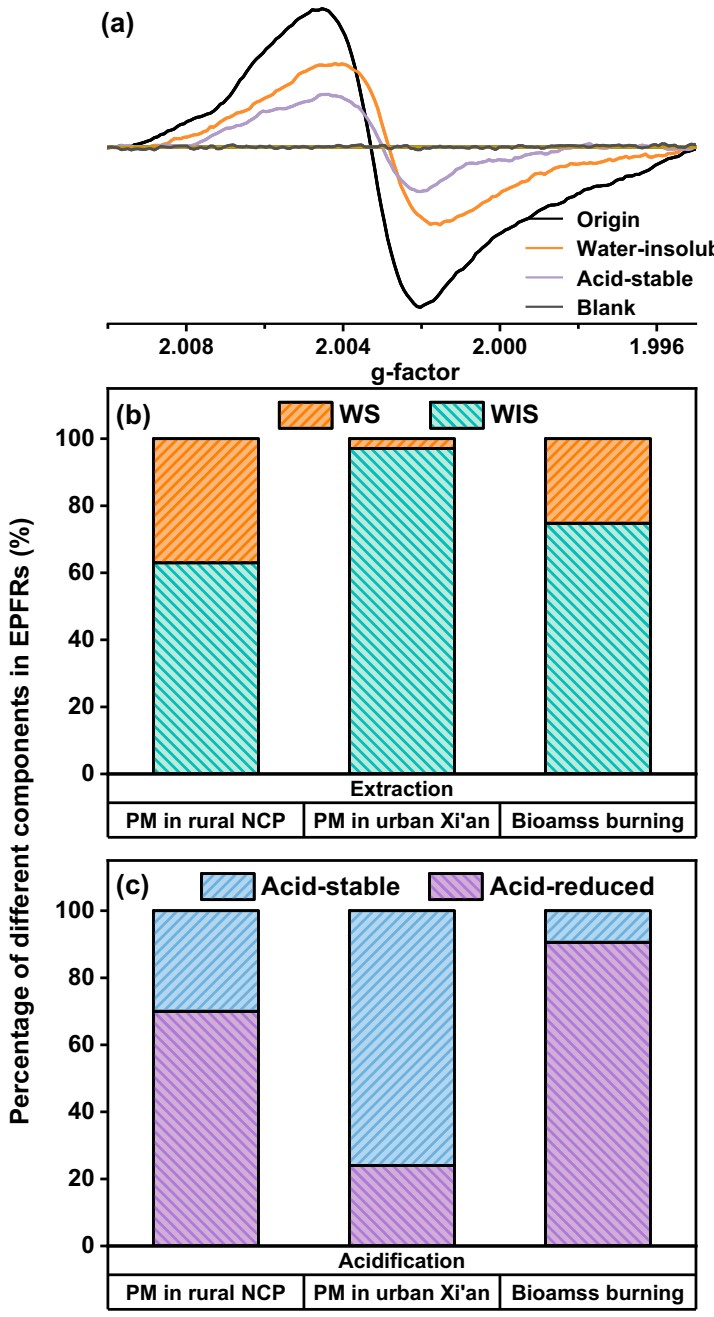

**Figure 5.** (a) Gaussian fitting EPR spectra of a selected PM$_{2.5}$ filter sample (sampling date: 2023/03/12) before (black) and after water extraction (orange), as well as after acidification (green). A typical EPR spectrum of a blank filter (grey) is also present for reference. (b) The percentages of different fractionated EPFRs in rural NCP (this study), in urban PM of Xi'an (Chen et al., 2018b), and biomass burning particles (Guo et al., 2023).



Importantly, compared to total EPFRs, water-soluble EPFRs (WS-EPFRs) showed stronger correlations with WS-OP

(Figure 6) of PM$_{2.5}$. In addition, no significant correlation was observed between water-insoluble EPFRs (WIS-EPFRs) and

WS-OP (Figures S9 a and b). These results demonstrate our hypothesis that the significant correlations between EPFRs and

WS-OP were driven by the water-soluble fractions of EPFRs, while atmospheric oxidation processes had increased the water

solubility of EPFRs. On the other hand, the lack of significant correlation between WIS-EPFRs and WIS-OP (Figures S9 c and

d) could be attributed to the complex interplay of organics and metals affecting WIS-OP (Gao et al., 2020). Nonetheless, the

significant correlation between WS-EPFRs and WS-OP suggests that atmospheric oxidation processes may have contributed

to the increased water solubility of EPFRs, thereby affecting their roles in PM$_{2.5}$ oxidative potential.

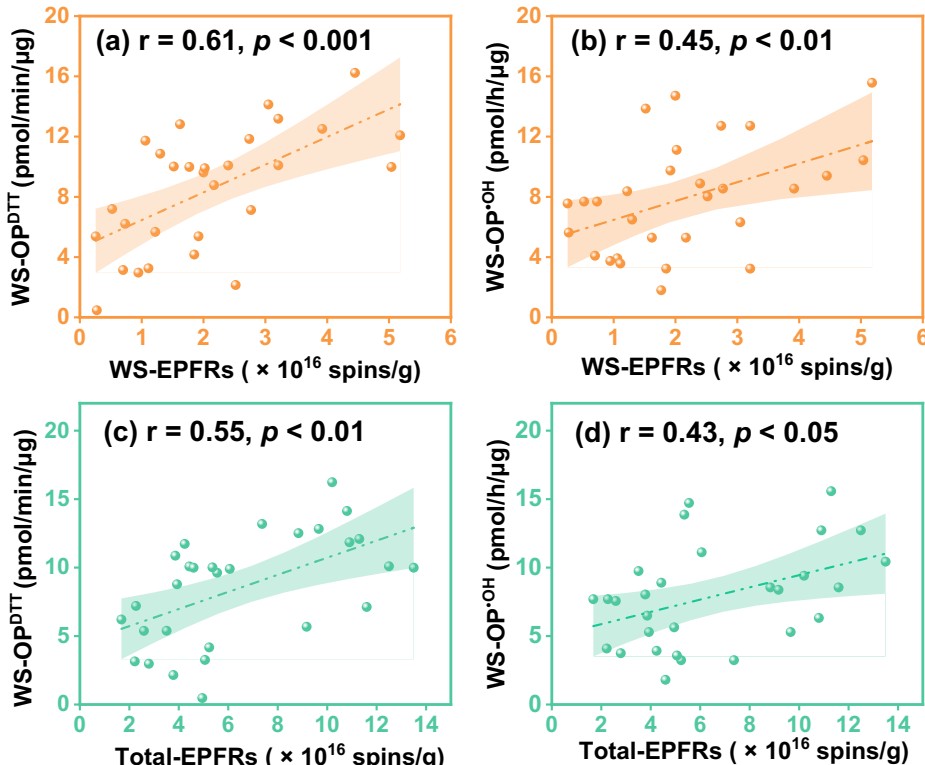

**Figure 6.** Correlation of WS/Total-EPFRs with WS-OP in $PM_{2.5}$; (a)WS-EPFRs with WS-OP$^{DTT}$ (b) WS-EPFRs with WS-OP$^{\bullet OH}$; (c) Total-EPFRs with WS-OP$^{DTT}$ (d)Total-EPFRs withWS-OP$^{\bullet OH}$. The Pearson correlation coefficients (r) and associated p values are illustrated in the figure. The lines and shadow areas are linear regressions with their 95% confidence intervals.

## 4. Conclusions and implications

In this study, EPFRs in fine, coarse, and total suspended particles in a typical rural region of the NCP have been investigated. The majority of EPFRs occurred in fine particles. EPFRs exhibited seasonal patterns distinct from those observed in urban environments. In addition, higher g-factors of EPFRs were identified compared to those reported for highway and urban PM, suggesting a greater extent of oxidation in rural EPFRs. These findings underscore a unique characteristic of EPFRs in rural areas, where local primary emissions of EPFRs are limited.

Additionally, atmospheric oxidation was resolved as the largest contributor to EPFRs. The atmospheric oxidation processes occurring during long-range/regional transport were suggested as the primary driver behind the more oxidized EPFRs observed in our work. We also demonstrated that the rural EPFRs contained a higher water-soluble fraction compared to those found in biomass burning particles and urban PM. Our results thus emphasize the importance of considering the water-soluble fraction of EPFRs, alongside their occurrence in organic solvent-extractable or non-extractable organics, particularly in regions without significant primary emissions of EPFRs. Future studies are warranted to investigate airborne EPFRs in other rural regions to yield a more complete understanding of the sources and properties of EPFRs, as only a typical rural site in NCP was examined in our work.

The WS-EPFRs could be an important contributor to the oxidative potential of $PM_{2.5}$, as suggested by their significant positive correlations. While prior research has predominantly focused on EPFRs in urban environments or originating from primary combustion emission sources, our findings revealed the evolution of EPFRs through atmospheric oxidation during transport (Figure 7). The atmospheric evolution of EPFRs may modify their properties, such as water solubility, thereby



altering their roles in contributing to the oxidative potential of PM (Wang et al., 2018c). It is also important to note that water-

soluble EPFRs are more bioavailable than their insoluble forms. They are more likely to reach the tissues or organs beyond

the lung deposition site, amplifying the threat to human health (Liu and Ng, 2023). Furthermore, beyond influencing the

oxidative toxicity of PM, the role of evolved EPFRs in climate-related cloud chemistry may also change and should be explored

in future research.

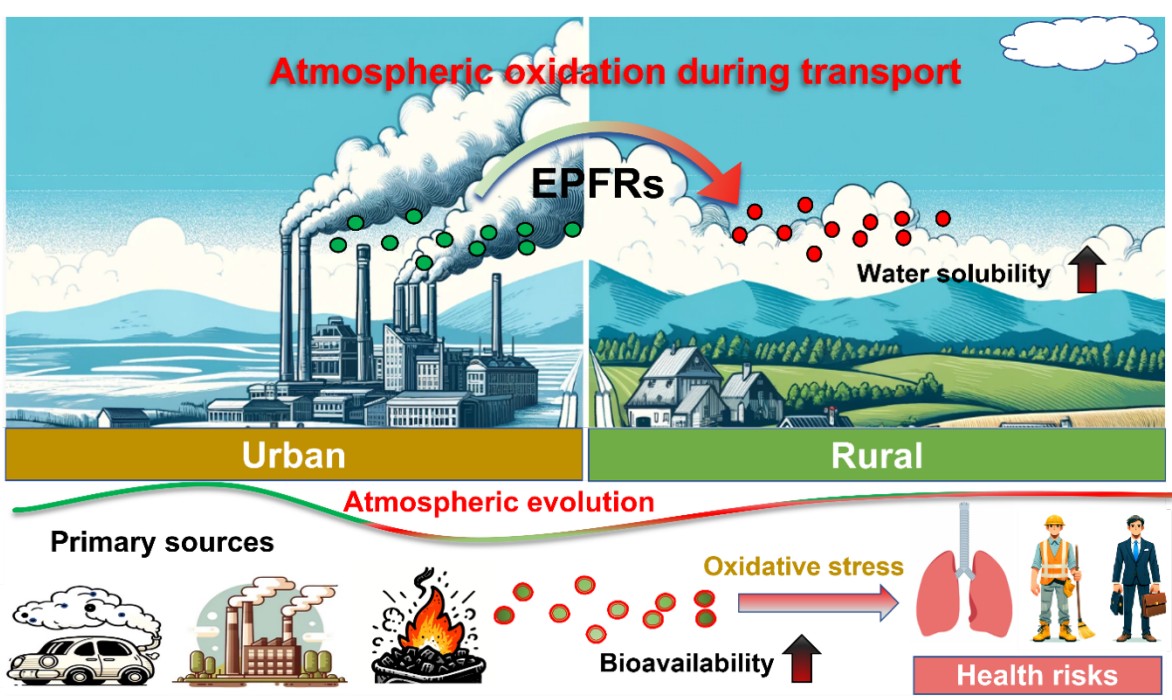

**Figure 7.** Implication of atmospheric evolution of EPFRs. EPFRs, primarily emitted from combustion sources, may undergo atmospheric oxidation during long-range/regional transport. This transformation may enhance the water solubility of EPFRs, potentially heightening their bioavailability and consequently elevating risks to human health.

Generally, soluble EPFRs are assumed to have a relatively short lifetime. For instance, it has been found that secondary

EPFRs formed by photoexcitation are highly susceptible to reduction by oxygen, leading to short lifetimes ranging from

approximately 30 minutes to one day (Chen et al., 2019). However, measurements conducted on several PM$_{2.5}$ filters stored at

-20 ℃ for roughly one year showed that EPR signals only decreased by approximately 11% (Figure S10) suggesting the

persistency of EPFRs, including their water-soluble fraction. Previous research has indicated that semiquinone-type radicals adsorbed into a polymeric carbonaceous core (Valavanidis et al., 2005) or by electron transfer with transition metals (Truong et al., 2010) are stable and can have a lifetime longer than months. However, their water solubility remains unclear. Further

efforts to identify the types of water-soluble EPFRs and investigate their lifetimes are warranted.

Overall, this study revealed the characteristics of EPFRs in rural regions, highlighting the importance of considering atmospheric oxidation processes in understanding their behavior and impacts. Future research efforts should aim to further elucidate the mechanisms governing the formation, evolution, and fate of EPFRs, as well as their interactions with other atmospheric components. Such insights are essential for developing effective strategies to mitigate the adverse effects of

385 EPFRs on air quality and public health, as well as for advancing our understanding of their broader implications for atmospheric and environmental science.

**Data availability**

The data are available upon request to the corresponding author Fobang Liu (fobang.liu@xjtu.edu.cn) and Haijie Tong (haijie.tong@hereon.de).

**Author contributions**

FL and XY designed the research. XY, SY, YY, YW, JL, MZ, and ZW carried out the experiments. FL, CH, HT, and KW supervised the study. FL and XY prepared the original manuscript with input from all the co-authors.

**Competing interests**

The authors declare that they have no conflict of interest.



## Disclaimer

Publisher's note: Copernicus Publications remains neutral with regard to jurisdictional claims made in the text, published maps, institutional affiliations, or any other geographical representation in this paper. While Copernicus Publications makes every effort to include appropriate place names, the final responsibility lies with the authors. Regarding the maps used in this paper, please note that Figure S1 contains disputed territories.

## Acknowledgments

This work is supported by the Natural Science Basic Research Program of Shaanxi (2023-JC-QN-0141), Qinchuanyuan introducing high-level innovation and entrepreneurship talent program (QCYRCXM-2022-363), and "Young Talent Support Plan" of Xi'an Jiaotong University (ND6J027). We gratefully acknowledge technical support from Professor Station of China Agricultural University at Xinzhou Center for Disease Control and Prevention. We acknowledge the use of ChatGPT 4.0 to generate the cartoons of Figure 7.

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
