# Peer review of "Atmospheric evolution of environmentally persistent free radicals in rural North China Plain: effects on water solubility and $PM_{2.5}$ oxidative potential"

_EGUsphere, 2024_

## Author Comment (AC1)

We thank the reviewers for their helpful comments. Our point-by-point response can be found below. The reviewers' comments are in *italics* and changes made to the manuscript are in blue. The line number mentioned corresponds to the tracked-change version. All the changes made do not affect the conclusions in the manuscript.

***Reviewer #1***

*The study examined the concentration, size distribution, and seasonal variations of EPFRs in the North China Plain region, as well as investigated their sources using PMF. It also explored the role of EPFR speciation in contributing to the oxidative potential of PM. I find this study very interesting and important. The authors have done a great job to discuss other studies comprehensively. I only have two comments as follows:*

> **Reply:** We thank the reviewer for the positive review and helpful comments. We have addressed each comment below.

1. *The conclusion that the majority of EPFRs are present in $PM_{2.5}$ is drawn from the fact that the average EPFRv in $PM_{2.5}$ accounts for over 95.2% of those in PM10 and TSP. However, the box plots in Fig. 1 suggest that $PM_{2.5}$ EPFRs may not make up such a high fraction. I recommend that the authors calculate the fraction of EPFRs in $PM_{2.5}$ in $PM_{10}$ or TSP for each sample and average it for discussion, for better representativeness.*

> **Reply:** We thank the reviewer for the suggestion. As noted in Line 86 of the original manuscript (now Line 87 of the revised manuscript), 24-hr $PM_{2.5}$, $PM_{10}$, and TSP samples were collected sequentially. Thus, calculating the fraction of EPFRs in $PM_{2.5}$/PM10/TSP for each sample was unable to perform in this study. However, we have examined the fraction of EPFRs in $PM_{2.5}$/PM10/TSP for each season in the revised manuscript. The results indicate EPFRv in $PM_{2.5}$ were all greater than 89.5% of those in PM10 and TSP, further suggesting that the majority of EPFRs are present in $PM_{2.5}$.
>
> Line 164: Similar results were found for EPFRv and PM concentrations in each season (Figure S5).

[Figure]

**Figure S5.** The concentrations of EPFRv (a) and PM (b) in different particle sizes in each season. The bars represent the standard deviations.

2. *In Section 3.3.2 and Fig. 4, the associations of OP with various chemical species are discussed. The OP values are in mass-normalized activities, while the chemical species are in ambient concentrations in m3. It would be helpful for the authors to explain why volume-*

*normalized OP data were not utilized for the association discussion, which makes more sense to me.*

**Reply**: In the revised manuscript, we have clarified in the text and the caption of Figure 4 that data of mass-normalized OP and mass fractions of chemical species were used for the correlation analyses. Mass-normalized OP represents the intrinsic redox properties of PM generated by chemical components. Volume-normalized OP is related to the actual exposure of the human body to redox-active substances. This work aims to identify individual chemical species influencing the intrinsic redox activity of ambient PM Therefore, correlation analyses between mass-normalized OP and mass fractions of chemical species were performed. We have clarified this point in the revised manuscript. Nevertheless, we have also included the correlation results between volume-normalized OP and chemical species per cubic meter of air in Figure S9 in the revised manuscript.

Line 283: To identify individual chemical species influencing the intrinsic redox activity of ambient PM, correlation analyses between mass-normalized OP ($OP^{DTT/\cdot OH}$) and the mass fraction of determined chemical species, were performed. The results are shown in Figure 4 and the detailed information is also listed in Tables S5–7 (the volume-normalized correlation results are also included in Figure S9 in case the readers are interested).

Line 305: **Figure 4.** Correlation coefficients (Pearson's r) of mass-normalized OP (Total/WS/WIS) with mass fractions of selected chemical species.

**Figure S9.** Correlation coefficients (Pearson's r) of volume-normalized OP (Total/WS/WIS) with selected chemical species per cubic meter of air.

---

## Author Comment (AC2)

We thank the reviewers for their helpful comments. Our point-by-point response can be found below. The reviewers' comments are in *italics* and changes made to the manuscript are in blue. The line number mentioned corresponds to the tracked-change version. All the changes made do not affect the conclusions in the manuscript.

*Reviewer #2*

*This study presents measurements of environmentally persistent free radicals (EPFRs) and oxidative potential (OP) of different size fractions of PM (PM2.5, PM10, TSP) collected over 1 year in a rural site in the North China Plane. The authors investigated the sources of EPFR using positive matrix factorisation and explored the role of solubility on EPFRs and their contribution to OP. Overall, the study is well written and presented, providing new insights into the sources contributing to EPFR, as well as the influence of EPFRs on OP, and the results are comprehensively compared to the literature. The findings in this study are novel and certainly within the scope of ACP. I recommend publication and only have a few minor comments.*

   **Reply:** We thank the reviewer for the positive review and helpful comments.

1. *Line 57 - I am unclear as to the meaning of nonsolvent-extractable, please clarify.*

   **Reply:** We have clarified the meaning of nonsolvent-extractable in the revised manuscript. It indicates the organic matter was unable to be extracted by water, methanol, dichloromethane, and n-hexane (Chen et al., 2018).

   Line 57: Notably, Chen et al. (2018) revealed that the dominant fraction of EPFRs existed within nonsolvent-extractable (unable to be extracted by water, methanol, dichloromethane, and n-hexane) organic matter of urban $PM_{2.5}$, underscoring the need for further exploration into the organic molecules associated with ambient EPFRs.

*2. Were filters in this study all extracted and analysed at the end of the sampling campaign or were they systematically analysed during the measurement campaign? This is important to clarify given the broad range of lifetimes of EPFRs.*

**Reply:** The EPFR measurements were conducted at the end of the sampling campaign. We have clarified the measurement time of EPFRs in the revised manuscript. However, as discussed in Line 375 of the original manuscript (now Line 382 of the revised manuscript), the EPR signals were similar for some filter samples after one year of storage at -20 ℃. Therefore, the influence of measurement time on EPFR concentrations should be minor in our work.

Line 109: All the EPFR measurements were conducted within one year after sampling.

*3. References for the DTT protocol should be provided.*

**Reply:** We have added the references for the DTT assay (and also the OH assay) in the revised manuscript.

Line 124: … yielding ultraviolet-detectable 2-nitro-5-thiobenzoic acid (Verma et al., 2012; Fang et al., 2015).

Line 128: …and the production rate of •OH was calculated based on the produced 2-OHTA, as the formation of 2-OHTA is proportional to the generation of •OH (Yu et al., 2022; Li et al., 2019).

*4. Line 121 – Typo "The rest DTT..."*

**Reply:** This has been corrected.

Line 123**:** The remaining DTT after the incubation was quantified by its reaction ....

*5. Line 122-126. The authors state that 2-OHTA production is proportional to OH formation, however Gonzalez et al (2018) demonstrated that the yield of the 2-OHTA from the terephthalate-OH reaction is around 31%, and thus calculated OH concentrations need to be*

*corrected concerning the yield of 2-OHTA . Have the author's considered this when calculation OPOH? Ref: https://doi.org/10.1080/00032719.2018.1431246*

**Reply:** Yes, we have considered the yield of 2-OHTA in calculating the OH concentrations. As suggested by previous studies (Li et al., 2019; Yu et al., 2022), a 35% yield of 2-OHTA for terephthalate-OH reaction in phosphate-buffered saline was used in this work. The detailed calculation method for OH concentration has been illustrated in the original and revised Supporting Information.

*6.     Regarding the correlation of OP to components (Section 3.3.2, Figure 4), mass normalised OP values were used, but were these correlated with mass normalised components (e.g. Fe ug-1)? This should be clarified. It also seems strange that correlation analysis between EPFR and components is not presented in the main manuscript in Figure 4, given a lot of the manuscript focus is on EPFRs.*

**Reply:** In the revised manuscript, we have clarified in the text and the caption of Figure 4 that data of mass-normalized OP and mass fractions of chemical species were used for the correlation analyses. Figure 4 is specifically to display the correlation results between OP and chemical species; therefore, correlations between EPFRs and chemical species are not included in the figure. However, the correlation results between EPFRs and chemical species have been provided in Table S10 of the original Supporting Information and brought up when discussing the EPFR formation in Line 315 of the original manuscript (now also in Table S10 and Line 321 of the revised manuscript).

**References**

Chen, Q. C., Sun, H. Y., Wang, M. M., Mu, Z., Wang, Y. Q., Li, Y. G., Wang, Y. S., Zhang, L. X., and Zhang, Z. M.: Dominant Fraction of EPFRs from Nonsolvent-Extractable Organic Matter in Fine Particulates over Xi'an, China, Environ. Sci. Technol., 52, 9646-9655, https://doi.org/10.1021/acs.est.8b01980, 2018.

Fang, T., Verma, V., Guo, H., King, L. E., and Edgerton, E. S.: A semi-automated system for quantifying the oxidative potential of ambient particles in aqueous extracts using the dithiothreitol (DTT) assay: results from the Southeastern Center for Air Pollution and

Epidemiology (SCAPE), Atmos. Meas. Tech., 8, 471-482, https://doi.org/10.5194/amt-8-471-2015, 2015.

Li, X. Y., Kuang, X. B. M., Yan, C. Q., Ma, S. X., Paulson, S. E., Zhu, T., Zhang, Y. H., and Zheng, M.: Oxidative Potential by PM2.5 in the North China Plain: Generation of Hydroxyl Radical, Environ. Sci. Technol., 53, 512-520, https://doi.org/10.1021/acs.est.8b05253, 2019.

Verma, V., Rico-Martinez, R., Kotra, N., King, L., Liu, J. M., Snell, T. W., and Weber, R. J.: Contribution of Water-Soluble and Insoluble Components and Their Hydrophobic/Hydrophilic Subfractions to the Reactive Oxygen Species-Generating Potential of Fine Ambient Aerosols, Environ. Sci. Technol., 46, 11384-11392, https://doi.org/10.1021/es302484r, 2012.

Yu, Q., Chen, J., Qin, W. H., Ahmad, M., Zhang, Y. P., Sun, Y. W., Xin, K., and Ai, J.: Oxidative potential associated with water-soluble components of PM2.5 in Beijing: The important role of anthropogenic organic aerosols, J. Hazard. Mater., 433, https://doi.org/10.1016/j.jhazmat.2022.128839, 2022.